

# Aerosol Properties and Their Influences on Low Warm Clouds during the Two-Column Aerosol Project

Jianjun Liu[1, 2] and Zhanqing Li[2, 3]

[1]Laboratory of Environmental Model & Data Optima, Laurel, MD, USA
[2]Earth System Science Interdisciplinary Center, University of Maryland, College Park, MD, USA.
[3]State Laboratory of Earth Surface Process and Resource Ecology, College of Global Change and Earth System Science, Beijing Normal University, Beijing, China.

*Correspondence to*: Jianjun Liu (jianjun5212@163.com), Zhanqing Li (zhanqing@umd.edu)

**Abstract.** Twelve months of measurements collected during the Two-Column Aerosol Project field campaign over Cape Cod, Massachusetts, which started in the summer of 2012, were used to investigate aerosol physical, optical, and chemical properties, and their influence on the
dependence of cloud development on thermodynamic (lower tropospheric stability, LTS) conditions. Relationships between aerosol loading and cloud properties under different dominant air-mass conditions and the magnitude of the first indirect effect (FIE), as well as the sensitivity of the FIE to different aerosol compositions, are examined. The seasonal variation in aerosol number concentration ($N_a$) was not consistent with variations in aerosol optical properties
(scattering coefficient, $\sigma_s$, and columnar aerosol optical depth), which suggests that a greater number of smaller particles with less optical sensitivity were present. Strong surface winds generally resulted in smaller $\sigma_s$ and a smaller contribution of fine particles to the total scattering extinction, but resulted in large $N_a$, suggesting that strong surface winds transported more aerosols with small particle sizes and less optical sensitivity to the site. The large contribution of
organics to small particle sizes was observed which decreased during the particle growth period. For low aerosol loading conditions, the liquid water path (LWP) and droplet effective radius (DER) significantly increase with increasing LTS, but for high aerosol loading conditions, LWP and DER changed little, indicating that aerosols significantly weaken the dependence of cloud development on LTS. The reduction in LWP and DER from low to high aerosol loading
conditions was greater in stable environments, suggesting that clouds in a stable conditions are more influenced by aerosol perturbations than those in more unstable conditions. High aerosol



loading weakened the increase in DER as LWP increased and strengthened the increase in COD with increasing LWP, resulting in the changes in the pattern of cloud properties each other. Under both continental and marine air-mass conditions, high aerosol loading can significantly made the shift in COD towards larger values, and in LWP and DER towards smaller values, and significantly narrowed the distribution of LWP and DER. Magnitudes of the FIE estimated under continental air-mass condition ranged from 0.07±0.03 to 0.26±0.09 with a mean value of 0.16±0.03 and showed an increase trend as LWP increased. The calculated FIE values for aerosols with a low mass of organics dominated cases are larger than that for aerosols with a high mass of organics dominated cases, implying that clouds over regions dominated by aerosol particles containing mostly inorganics are more susceptible to aerosol perturbations, resulting in larger climate forcing, than clouds over regions dominated by aerosol particles containing mainly organics.

## 1 Introduction

Aerosols can significantly influence climate change through their direct and indirect effects. The aerosol direct effect is when aerosol particles change earth's radiative balance by scattering and absorbing solar radiation. The aerosol indirect effect is when aerosols change cloud microphysical, macrophysical, and precipitation properties through their role as cloud condensation nuclei (CCN) or ice nuclei (IN). Under constant liquid water path (LWP) conditions, an increase in aerosol concentration will lead to an increase in CCN concentration. This results in an increase in cloud droplet number concentration, a decrease in cloud droplet effective radius (DER), and a more reflective cloud. This is referred to as the first aerosol indirect effect (FIE) (Twomey, 1977). The decrease in DER will reduce the chances of precipitation forming, which prolongs the lifetime of a cloud and enhances its LWP. This is known as the second aerosol indirect effect. Estimates of indirect aerosol effects have large uncertainties (Boney and Dufresne, 2006; Lohmann et al., 2010). This makes the impact of aerosols on the prediction of the current and future behavior of earth's climate system highly uncertain (McComiskey et al., 2008; IPCC, 2013).

The observed response of warm low cloud properties to aerosol properties has been observed from satellite-based remote sensing (Bréon et al., 2002; Lebsock et al., 2008; Su et al., 2010; Wang et al., 2014), surface-based remote sensing (Kim et al., 2003; Feingold et al., 2003; Feingold et al., 2006; McComiskey et al., 2009; Liu et al., 2016; Liu et al., 2018a), combined



surface measurements and satellite retrievals (Sporre et al., 2012, 2014), as well as by aircraft measurements (Zhang et al., 2011; Twohy et al., 2013; Painemal and Zuidema, 2013; Werner et al., 2014). Most of the above studies have shown that DER significantly decreases with increases in aerosol loading. However, LWP can increase or decrease with aerosol loading, depending on

cloud thermodynamics and dynamics (Han et al., 2002). Current estimates of FIE from all available observational platforms have a large range of values because each measurement approach has its own set of uncertainties. The large uncertainty and range in estimates of the FIE results in a large uncertainty in aerosol indirect radiative forcing (McComiskey and Feingold, 2008). Narrowing uncertainties in measures of aerosol-cloud interactions (ACI) and developing

well-constrained parameterizations for models requires analyses of ACI over different climatic and aerosol regions of the Earth.

      Cloud development is significantly influenced by large-scale thermodynamic conditions, such as lower tropospheric stability (LTS). Changes in ACI as LTS changes have been widely investigated using observations made from the surface and from satellite remote sensing (Matsui

et al., 2004; Su et al., 2010; Liu et al., 2016). However, to what degree the dependence of cloud development to aerosol perturbations are related to large-scale thermodynamic conditions is not well known. Moreover, the mechanism behind the aerosol FIE is that aerosols affect the cloud droplet number and the cloud droplet effective radius (DER) through their role as CCN, which is determined by the aerosol particle size, number concentration, and chemical composition

(Menon et al., 2002; Sekiguchi et al., 2003; Wang et al., 2008). A significant influence of aerosol hygroscopicity on the magnitude of the aerosol FIE when aerosol optical quantities are used to estimate the FIE was reported (Liu et al., 2018b). The role of aerosol size and number concentrations on the FIE has been examined (Komppula et al., 2005; Anttila et al., 2009), whereas the question of how sensitive cloud properties to aerosol composition in addition to

aerosol loading is still under investigation (Hao et al., 2013; Portin et al., 2014).

      The Department of Energy's Atmospheric Radiation Measurement (ARM) mobile facility was stationed at Cape Cod, Massachusetts from July 2012 to June 2013 for the Two-Column Aerosol Project (TCAP) field campaign (Berg et al., 2016). Measurements of aerosol, radiation, and cloud characteristics were made at the site which is subject to both clear and cloudy

conditions as well as clean and polluted conditions. The site is commonly influenced by different air-masses, such as the continental, marine and continental-marine mixed air mass. This study uses the data collected during the TCAP field campaign to investigate aerosol physical, optical,



and chemical properties, and their influence on the dependence of cloud development on large-scale thermodynamic conditions under different air-masses influenced. The influence of aerosol loading on cloud properties under different air mass conditions and the magnitude of the FIE, as well as the question of how sensitivity of the FIE to different aerosol compositions in addition to aerosol loading, are also investigated. Data and methods used in this study are described in section 2. Seasonal variations in aerosol physical, optical, and chemical properties and their influence on low warm clouds are presented in section 3. Conclusions are given in section 4.

## 2    Data and methods

### 2.1    Aerosol properties

#### 2.1.1    Surface aerosol properties

The optical properties of surface aerosols were measured by instruments making up the Aerosol Observation System (AOS), which is the primary ARM platform for in situ aerosol observations. The TSI-3010 condensation particle counter was used to obtain the total number concentration of condensation particles ($N_a$) with diameters greater than 10 nm. The scattering ($\sigma_s$) and absorption ($\sigma_a$) coefficients of total ($\leq 10$ μm) and fine mode ($\leq 1$ μm) aerosol particles were measured under dry conditions with a relative humidity (RH) level equal to 40% using a TSI-3653 nephelometer at three wavelength (450, 550, and 700 nm) and a Radiance Research particle soot absorption photometer (PSAP; 470, 528, and 660 nm), respectively (Jefferson, 2011). Nephelometer and PSAP measurements have been calibrated and quality controlled using the methods developed by Anderson and Ogren (1998) and Anderson et al. (1999), respectively. Measurements of $\sigma_a$ at 470 nm were normalized to 450 nm to match $\sigma_s$ measurements. The single scattering albedo (SSA) of surface aerosol particles is then calculated as $\sigma_s/(\sigma_s + \sigma_a)$ using $\sigma_s$ and $\sigma_a$ at 450 nm.

The aerosol size distribution ranging from 15 nm to 450 nm was measured by a scanning mobility particle sizer (SMPS) with five-minute averaging. The SMPS contains a cylindrical differential mobility analyzer (Model 3081) and a TSI (Model 3010) and was calibrated using polystyrene latex standards (Wang et al., 2003). The bulk chemical composition of the non-refractory components of sub-micron (aerodynamic diameter = ~40-1000 nm) aerosol particles (organics, sulfate, nitrate, ammonium, and chloride) was measured by an aerosol chemical speciation monitor (ACSM), which is a thermal vaporization, electron impact ionization mass spectrometer build upon the same technology as the widely used aerosol mass spectrometer.



Under ambient conditions, the detect limitation of mass concentrations of particles is less than 0.2 µg/m$^3$ for 30-minute signal averaging. The ACSM is calibrated with ammonium nitrate following the method of Ng et al. (2011).

### 2.1.2  Columnar aerosol properties

5  Columnar aerosol optical depths (AOD) and Angstrom exponents (AE) were obtained from the Aerosol Robotic Network (AERONET) database (Holben et al., 1998). AODs are retrieved from direct Sun measurements with an uncertainty of 0.01-0.02 (Eck et al., 1999). In this study, Level 2.0 quality-assured and cloud-screened data was used.

## 2.2  Cloud properties

### 2.2.1  Cloud boundaries

10  Cloud-base and cloud-top heights were identified using a combination of observations from the 95 GHz W-band ARM cloud radar (WACR), the micropulse lidar (MPL), and the ceilometer (Kollias et al., 2007). The algorithm used in the cloud boundary retrieval is similar to the method developed by Clothiaux et al. (2000), which is based on 35-GHz millimeter cloud radar observations. The WACR cloud and precipitation mask is derived from signal-to-noise ratio thresholds determined for each time profile. An MPL cloud mask is combined with ceilometer cloud-base estimates to produce a best-estimate cloud-base for each time point. The MPL and WACR cloud masks are merged, and then additional filtering of the resulting cloud mask is done in the lower troposphere to remove insect returns. Insects are identified using a combination of WACR linear depolarization ratio and reflectivity measurements. The temporal and vertical resolution of the cloud boundary product is 5 seconds and 42.856 m, respectively.

### 2.2.2  Cloud microphysical properties

Cloud optical depths (COD) and liquid water paths (LWP) were retrieved based on measurements from a two-channel narrow field-of-view (NFOV) radiometer and a microwave radiometer profiler (MWRP). The cloud droplet effective radius ($r_e$) was calculated using the following equation:

$$\tau = \frac{3LWP}{2\rho_w r_e}$$

where $\rho_w$ is the density of water. The NFOV radiometer with a 5.7° field-of-view measuring downwelling zenith radiances at 673 nm and 870 nm, which is used to retrieved COD using the



method described by Chiu et al. (2010) and Liu et al. (2013). Simultaneous radiance measurements with a high accuracy from the AERONET Sun photometer (Holben et al., 1998) were used to quantify biases in the NFOV radiance measurements (Fig. 1). AERONET and NFOV radiances agree well at 673 and 870 nm (R=0.99 in both cases). However, NFOV-measured zenith radiances at 673 nm are underestimated by ~15%. Consequently, NFOV measurements at 673 nm were adjusted using following formula:

$$F_{673,adj} = 1.1519 * F_{673,obs} + 0.0007$$

where $F_{673,obs}$ represents measured zenith radiances and $F_{673,adj}$ represents adjusted radiances at 673 nm. The total uncertainty in COD retrievals using this method is ~17% (Chiu et al., 2010). The MWRP built by the Radiometrics Corporation measures atmospheric brightness temperatures at 12 frequencies. LWPs were retrieved using brightness temperatures measured at the five K-band channels (22.235, 23.035, 23.835, 26.235, and 30.0 GHz) based on a statistical retrieval algorithm developed by Liljegren et al. (2004). The typical uncertainty in LWP retrievals from microwave radiometers is ~20 g m$^{-2}$ for LWP < 200 g m$^{-2}$ and ~10% for LWP > 200 g m$^{-2}$ (Dong et al., 2008; Liljegren et al., 2004).

In this study, only non-precipitation, low warm clouds with cloud top height less than 3 km are considered. The LWP observations less than 40 g m$^{-2}$ and greater than 300 g m$^{-2}$ were excluded to avoid very thin or broken cloud cover, as well as post-precipitation conditions (McComiskey et al., 2009) and potential precipitation contamination (Dong et al., 2008).

**2.3    Surface and large-scale meteorological parameters**

Surface meteorological parameters during the campaign period were measured by the ARM surface meteorological system at a 1-minute resolution. The large-scale vertical motion (ω) at 700 hPa and LTS is used in this study to constrain large-scale dynamic and thermodynamic conditions (Su et al., 2010; Medeiros and Stevens, 2011; Liu et al., 2016). LTS is calculated as the difference between the potential temperature of the free troposphere (700 hPa) and the surface. Values of ω and potential temperature were obtained from the European Centre for Medium Range Weather Forecasts model runs for ARM analysis with a one-hour resolution for a 0.56º x 0.56º box centered on the site.

**2.4    Air-mass trajectories classification**



Two-day air mass back trajectories arriving at the site at 500 m at midnight were simulated using the HYSPLIT model (Stein et al., 2015; Rolph, 2016), and all simulated trajectories are classified into three clusters. Cluster I represents the continental air-masses, which generally originated from continental area located at the west of the site and moved over the site. The air-masses which originated from ocean of east of site and directly moved to the site are identified the marine air-masses (cluster II). And the cluster III represents an air-mass that passed over both continental regions and the ocean to the site with anthropogenic and marine aerosol influenced. During the study period, the occurrence of cluster I, II and III air-masses was 62.5%, 15.9% and 21.6 %, respectively.

## 3    Results

### 3.1    Variations in aerosol properties

#### 3.1.1    Seasonal variations in aerosol optical properties and number concentration

Figure 2 show monthly statistics describing surface-measured $\sigma_s$ for total ($\sigma_{10}$) and fine mode ($\sigma_1$) aerosol particles and $N_a$. Seasonal and annual mean values are summarized in Table 1. Maxima in $\sigma_1$ and $\sigma_{10}$ are found in the summer months and minima in $\sigma_1$ and $\sigma_{10}$ are found in the winter months. Fine particles dominate aerosol scattering in the summertime and are responsible for ~75% of total particle scattering. The contribution of fine particle scattering to total particle scattering in other seasons ranges from ~46% to ~54%, indicating that particles with sizes ≤ 1 µm and 1-10 µm play a similar role in aerosol scattering extinction. Monthly and seasonal variations in $N_a$ show that maximum and minimum seasonal mean $N_a$ occurs in spring and autumn, respectively, which is not consistent with the variations in aerosol scattering coefficient. This inconsistency is probably due to the difference in aerosol particle size distribution in each season since aerosol extinction properties significantly depend on particle size. The largest values of $N_a$ corresponding to moderate values of $\sigma_s$ are found in spring and are likely due to the presence of a greater number of smaller particles with less optical sensitivity. The total particle SSA shows a slight seasonal variation, suggesting smaller changes in aerosol particle absorption properties. Figure 3 shows monthly statistics describing columnar AOD and AE. Seasonal and annual mean values are summarized in Table 1. The variation in AOD and AE is consistent with the variation in surface-measured $\sigma_s$ and the ratio $\sigma_1/\sigma_{10}$, indicating the surface aerosol properties can represent the columnar aerosol properties very well. Figure 4 shows monthly mean wind speeds and wind directions during the campaign period. Monthly mean wind speeds ranged from



~3.8 m/s to 6.6 m/s and southwesterly winds dominated throughout the whole year over the area. Months with the strongest mean surface wind speeds generally have small $\sigma_s$ with a small contribution of fine particles to total scattering extinction. However, relatively large aerosol number concentrations were measured. This indicates that strong surface wind speeds

transported smaller aerosol particles with no optical sensitivity from the continental interior to over the site.

### 3.1.2    Aerosol optical properties under different air-masses conditions

Table 2 gave the discrepancies in aerosol properties when different air-masses influenced on the site. The mean value of $\sigma_1$ is the largest/smallest under the continental/marine air-mass

conditions; however, $\sigma_{10}$ is the largest under cluster III condition and shows similar values under cluster I and II conditions. The inconsistent variations in $\sigma_1$ and $\sigma_{10}$ under different air-mass conditions are due to the particles with different size dominated, as indicated by $\sigma_1/\sigma_{10}$. When the continental air-mass influenced the site, the fine particles dominate aerosol scattering and are responsible for ~65% of total particle scattering, indicating the more anthropogenic aerosols with

small particle size are transported from continental regions. The values of $\sigma_1/\sigma_{10}$ under cluster II and III air-mass conditions show that the fine-mode and coarse-mode particles play the similar role on the total particle scattering. The variation in $N_a$ is consistent with that in $\sigma_1$ with the largest and smallest values under cluster I and II conditions, respectively. Smaller SSA values are found under continental air-mass conditions suggested the more absorbing aerosols are found

than that under other air-masses conditions due to the anthropogenic influence. The variation in AOD under each air-mass condition shows the similar values and AE is consistent with the variation in the ratio $\sigma_1/\sigma_{10}$.

### 3.1.3    Aerosol chemical composition and size distribution

Figure 5 shows the size distribution and the corresponding mass fraction of organics, sulfate,

ammonium, and nitrate of surface aerosol particles sampled in July and August 2012. New particle formation and growth periods were detected and are outlined by red rectangles in Figure 5. During the measurement period, fine particles containing more organics were dominant with a mean particle radius of 91.4±20.6 nm and a mean organic mass fraction of 0.67±0.16. Mean mass fractions of sulfate, ammonium, and nitrate are 0.18±0.11, 0.10±0.09, and 0.04±0.02,

respectively. At the beginning of new particle formation and growth periods, organics contribute the most to small particle sizes. Their contribution decreases as the growth period progressed to



be replaced by contributions from inorganics, in particular, sulfate. This is possible because sulfate ions are formed during nucleation involving neutral gaseous species like ammonia and sulfuric acid (Crilley et al., 2014). Small aerosol particles generally contribute more organics to the total aerosol mass over the study site, which can also be seen in the relation between mean

aerosol particle radii with organic mass fraction (Fig. 6). The strong decrease in aerosol particle size with increase in organic mass fraction has also been reported by others (Broekhuizen et al., 2006; McFiggans et al., 2006).

## 3.2   Aerosol, cloud, and meteorological conditions

### 3.2.1   Aerosol effects on the dependence of cloud properties on meteorological conditions

Low warm cloud properties are sensitive to changes in the thermodynamic conditions (Su et al., 2010; Medeiros and Stevens, 2011; Liu et al., 2016). Figure 7 shows cloud properties (LWP, and DER) as a function of LTS under low and high aerosol index (AI) conditions for continental and marine air-mass conditions. In this study, AI is used as the CCN proxy (Nakajima et al., 2001; Liu and Li, 2014), which is defined as the surface-measured aerosol scattering coefficients

multiple by surface-measured scattering angstrom exponents. Low and high AI are defined as the lowest and highest quarter of all AI samples, respectively (the same hereinafter). The differences in the meteorological parameters (such as temperature, wind speed, relative humidity etc.) at surface, and 850 hPa, large-scale dynamic ($\omega$) and thermodynamic parameters (LTS) are not significant for the low and high AI conditions (figure not shown). Table 2 summarized the mean

and standard deviation of cloud properties under each air-mass condition. Clouds influenced under the marine air mass conditions (cluster II) have the largest COD, LWP and DER (33.0±18.3 and 243±197 g m$^{-2}$, 10.9±6.6 μm respectively), and clouds associated with air mass from continental areas (cluster I) have the smallest cloud properties with COD of 25.7±14.5, LWP of 127±99 g m$^{-2}$ and DER of 7.9±4.8 μm. The top panels of Fig. 7 show that LWP

significantly increases with increasing LTS under low aerosol condition, which is consistent with those from studies using surface-based measurements (e.g. Liu et al., 2016) satellite measurements (e.g., Su et al. 2010), aircraft measurements (e.g., Cecchini et al. 2016), and model simulations as well (e.g., Johnson et al., 2004; West et al. 2014). The Johnson et al. (2004) simulations indicated that increasing in stability can induce the increases in the buoyancy of free-

tropospheric air above the temperature inversion capping the boundary later, inhibiting the entrainment of dry air through the cloud-top, resulting in the increases in LWP. Under high





aerosol conditions, LWP change little as LTS increases. The lower LWP under more aerosol conditions possibly because the inhibited cloud droplet sedimentation due to the reduced cloud droplet size likely enhances evaporation and entrainment at the cloud top, resulting in a reduction in LWP (Kaufman et al. 2005; Hill and Feingold 2009; Liu et al., 2016). Similar with the

variations in LWP with LTS, the DER under both air-masses conditions shows significant increases with increasing LTS under less polluted condition and slight changes with increasing LTS under high polluted condition. The changes in DER with LTS can possible because the changes in LWP with LTS due to the high positive correlation each other (Zhang et al., 2011; Sporre et al., 2014). The enhanced LWP under highly stable conditions can supply the water

needed for cloud droplet growth (Su et al. 2010; Zhang et al. 2011). And the increase in LWP is commonly accompanied by an increase in droplet collision–coalescence, resulting in the decrease in cloud number concentration, leading to an increase in DER (Kim et al. 2008; McComiskey et al. 2009; Liu et al., 2016). Differences in LWP and DER between low and high LTS conditions are larger under low pollution conditions than under high pollution conditions.

This suggests that high aerosol concentrations can significantly weaken the thermodynamic influence on the increase in LWP and DER due to the aerosol perturbation. The results imply that under the similar thermodynamic conditions, the development of clouds in a highly polluted environment is inhibited, which reduces the chances of precipitation because the rainfall frequency of warm low clouds is highly correlated with LWP (Chen et al., 2011; Liu et al., 2013).

Meanwhile, for all LTS bins, clouds under high aerosol conditions have lower values of LWP, and DER than clouds under low aerosol conditions. The reduction in LWP and DER is greater in stable environments than in unstable environments, suggesting that clouds in stable environments are more affected by the aerosol perturbation than those in more unstable regimes, which is consistent with the studies on marine warm clouds based on the surface measurements

(Liu et al., 2016).

### 3.2.2   Aerosol effects on the relationship among cloud properties

Figure 8 shows the dependence of COD and DER on LWP under low and high AI conditions. Under high AI conditions, COD increases sharply as LWP increases while under low AI conditions, COD changes little as LWP increases due to the decrease in DER influenced by

aerosol perturbation (Fig. 8a and b). Figure 8c and 8d suggests that the DER is sensitive to LWP. An increase in LWP leads to a significant increase in the size of cloud droplets (Zhang et al., 2011; Sporre et al., 2014). The increase in DER with LWP is more rapid under low AI conditions



than under high AI conditions. This is because there is a limit to the size a cloud droplet can reach when a given amount of water is shared among a large number of particles (Zhang et al., 2011). High aerosol loading conditions weaken the increase in DER and strengthen the increase in COD as LWP increases, indicating that aerosols can influence on the pattern of COD-LWP and DER-LWP.

Figures 8 also shows that across all LWP bins, COD is larger and DER is smaller under high AI conditions than under low AI conditions, which is consistent with the "Twomey" effect. The large differences between COD under low and high AI conditions at high LWP values (Fig. 8a and b) and between DER under low and high AI conditions at high LWP values (Fig. 8c and d) suggests that when clouds have large LWPs, aerosols will tend to inhibit the growth of cloud droplets more. This can happen because under high aerosol loading conditions, more aerosol particles are activated into CCN and cloud droplet concentrations will increase rapidly as LWP increases. However, under low aerosol loading conditions, cloud droplet concentrations increase slowly as LWP increases due to the lack of CCN source, so the size of cloud droplets increases rapidly as LWP increases (Zhang et al., 2011).

## 3.3  Aerosol effect on the cloud properties

### 3.3.1  Variations in cloud properties with aerosol loading under different air mass conditions

Probability distribution functions (PDFs) of COD, LWP, and DER under low and high AI conditions for air mass cluster I and II are shown in Fig. 9. Numbers written in each panel are the percentage differences in each cloud property defined as $(M_{ch} - M_{cl})/M_{cl} * 100\%$, where $M_c$ represents the mean value of a cloud property and subscripts $h$ and $l$ represent high and low AI conditions, respectively. The figure shows that the PDFs of COD, LWP and COD under high and low AI conditions differ significantly under both air-masses conditions. Although the peak value of COD is similar under low and high aerosol loading conditions, clouds under more polluted conditions have more large values of COD than that under less polluted conditions with enhancement in COD from low to high aerosol loading of 24.2% and 21.9% for cluster I and II air-mass, respectively. For the low aerosol loading case, the PDF of LWP shows a broad maximum with values between 50-180 g m$^{-2}$ and 80-230 g m$^{-2}$ for cluster I and II air-mass, respectively. The high aerosol loading cases, conversely, has a narrower PDF a distinct peak at 60-70 g m$^{-2}$. Under high AI conditions, the LWP decreases on the order of 30% and 45% from





their values under low AI conditions for cluster I and II air-mass, respectively. Under both air-masses conditions, there is a sharp shift in DER towards smaller values under high aerosol loading conditions than that under low aerosol loading conditions. Under low polluted conditions, the DER values show a broad range and generally higher values with most observations varying

between 5-12 µm for cluster I air-mass and peak around 15 µm for cluster II air-mass cases, respectively. For high polluted case, the PDF of DER for both air-masses conditions is significantly narrower and most of the values smaller than 10 µm with peak values around 5 µm. The large difference in DER under high and low aerosol loading conditions with value of ~40% and 55% are founded for cluster I and II air-mass, respectively. As indicated above, the

meteorological parameters and large-scale dynamic and thermodynamic parameters showed no significant differences for low and high AI conditions indicating the changes in cloud properties are mainly contributed to aerosols. Generally, clouds under marine air-mass conditions have slightly larger decreases in LWP and DER from low to high aerosol loading than those under continental air-mass conditions.

### 3.3.2   Aerosol first indirect effect

The aerosol FIE is generally quantified as

$$FIE = -\frac{dln(DER)}{dln(\alpha)}\Big|_{LWP},$$

where α represents CCN or CCN proxies. The FIE represents the relative change in mean cloud DER with respect to a relative change in aerosol loading for clouds having the same LWP

(Feingold et al., 2003). In the studies, the aerosol index is used as the CCN proxy (Nakajima et al., 2001; Liu and Li, 2014). Cloud samples were categorized according to their LWP values. The LWP bins range from 40-200 g m$^{-2}$ in increments of 20 g m$^{-2}$. The choice of a small increment ensures that the LWP constraint is met in each bin. Due to the lack of samples for cluster II air-mass condition, FIE is calculated only for clouds and aerosols under cluster I air-

mass condition. And only the values of FIE are statistically significant at the 95% confidence level (P=0.05) are discussed in the study. Figure 10a shows DER as function of AI for clouds with LWP ranging from 120-140 g m$^{-2}$ as an example to explain how to estimate the FIE in the study. The significant decrease in DER with increase in AI is founded. For this case, the magnitude of the FIE is 0.26 with an uncertainty of 0.09. The magnitudes and uncertainties of

FIE calculated in each LWP bin are shown in Fig. 10b. Numbers above each bar are the number of samples that went into the calculation of the FIE in each LWP bin. The magnitude of the FIE





changes from 0.07±0.03 to 0.26±0.09 with the smallest value found in the LWP bin of 40-60 g m$^{-2}$ and the largest value found in the LWP bin of 120-140 g m$^{-2}$. The mean value of FIE for all LWP bins is 0.16±0.06 during the study period. The values of FIE in each LWP shows an obvious increase with increasing of LWP, especially for LWP smaller than 140 g m$^{-2}$. This is

consistent with some of previous studies (e.g. Pandithurai et al., 2009; Sporre et al., 2014; Harikishan et al., 2016), can is possibly because the aerosol activation is enhanced due to increase of LWP (Zhao et al., 2012; Painemal and Zuidema, 2013). At higher LWP, with the availability of more CCN, more droplets can get activated. The droplet number increases, but their size decreases at fixed LWP (Harikishan et al., 2016). Estimates of the FIE reported from

all available platforms range widely and are sensitive to the definition of the aerosol burden (Lihavainen et al., 2010), the methods for retrieving cloud properties (McComiskey et al., 2009), and meteorological conditions, such as vertical velocity and atmospheric stability (Feingold et al., 2003; Matsui et al., 2004; McComiskey et al. 2009; Liu et al., 2016). Theoretical values of the FIE lie between 0 and 0.33 (McComiskey and Feingold, 2008) with most values falling between

0.05 and 0.25 (Zhao et al., 2012). Based on the surface retrievals, Feingold et al. (2003) derived FIE values of 0.02-0.16 with a mean value of 0.10±0.05 for a set of seven cases, and Kim et al. (2008) found that FIE values ranged from 0.04 to 0.17 at five LWP bins with a mean value of 0.09±0.05 from a 3-year (1999-2001) study at South Great Plain (SGP) in U.S., respectively. The mean FIE value of 0.07±0.01 for warm marine boundary clouds at Azores (Liu et al., 2016),

0.14±0.09 for continental clouds during monsoon period at a rural continental site over Mahabubnagar, India (Harikishan et al., 2016), and a range of 0.05-0.16 over the coastal region at Pt. Reyes, California (McComiskey et al., 2009) are reported based on the surface-based retrievals. The magnitude of the FIE in this study generally falls in this range.

To examine the question of how sensitive are cloud properties to aerosol composition in

addition to aerosol loading, the sensitivity of cloud properties to aerosol chemical composition represented by the mass fraction of organics was examined. The aerosol number concentrations are used as CCN proxy (Li et al., 2011; Yan et al., 2014; Liu et al., 2016) here due to the limitation of aerosol scattering coefficient measurements during the aerosol chemical composition observation period. Three LWP bins were defined: 40-60 g m$^{-2}$, 60-80 g m$^{-2}$, and

80-100 g m$^{-2}$. DER as a function of $N_a$ in each LWP bin when aerosol particle mass fractions of organics are low and high are shown in Fig. 11. Aerosols with low and high mass fractions of organics are defined as aerosols with mass fractions of organics smaller than and greater than,





respectively, the mean value of mass fraction of organics of all samples in each LWP bin. Mean values of ω and LTS in each aerosol particle mass fraction of organics category are given in the figure. Differences in ω and LTS between low and high mass fraction of organics are not significant in any LWP bin. Estimates of the FIE when aerosol samples with low mass fractions

of organics dominate are 0.10±0.05, 0.15±0.06, and 0.23±0.12 (see Fig.11a-c, respectively), which are greater than the estimates of the FIE when aerosol samples with high mass fractions of organics dominate (0.07±0.04, 0.12±0.06, and 0.07±0.05, respectively). This suggests that clouds under a majority of aerosol particles composed of inorganic compounds conditions are more susceptible to aerosol perturbations, resulting in a greater climate forcing, than clouds under a

majority of aerosol particles composed of organic compounds conditions. The mechanism behind the AIE is characterized by the ability of aerosol particles to act as CCN, which is primarily governed by particle size and chemical composition (McFiggans et al., 2006). The cloud-nucleating ability of aerosol particles is significantly greater when aerosol particles are large and are composed of more inorganic compounds that when they are small and are composed of more

organic compounds (Dusek et al. 2006; Liu et al., 2011). This study (figure 5) and others have demonstrated that aerosols containing more organic particles are generally smaller than those with more inorganic particles (Broekhuizen et al., 2006; McFiggans et al., 2006; Zhang et al., 2011), and organic particles are generally less CCN-active than inorganic particles (Raymond and Pandis, 2002; Zhang et al., 2011). This can partly explain the smaller FIE values induced by

aerosols with large mass fractions of organics.

## 4    Conclusions

Twelve months (July 2012 – June 2013) of measurements of aerosol and cloud properties, as well as meteorological conditions were collected during the Two-Column Aerosol Project (TCAP) field campaign over Cape Cod, Massachusetts. The goal of this study is to characterize

aerosol physical, optical, and chemical composition properties, and to determine their influence on cloud properties and the dependence of cloud development on large-scale thermodynamic conditions. The magnitude of the aerosol first indirect effect (FIE) and the question of how sensitive are cloud properties to aerosol composition in addition to aerosol loading were also examined.

The maximum and minimum in $\sigma_1$ and $\sigma_{10}$ were found in summer and winter, respectively. Fine particles dominated aerosol scattering in the summer and contributed toward ~75% of the total particle scattering. In other seasons, fine particles contributed toward ~45-54% of the total





particle scattering. The maximum and minimum mean values of $N_a$ occurred in spring and autumn, which is not consistent with the variation in aerosol scattering coefficient ($\sigma_s$). The variation in aerosol optical depth (AOD) is consistent with the variation in surface-measured $\sigma_s$ and inconsistent with the variation in $N_a$. This suggests that a large number of particles with less

optical sensitivity were present. Months with strong mean surface wind speeds were generally associated with small $\sigma_s$ and a small contribution of fine particles to total scattering extinction, but relatively large aerosol number concentrations. This suggests that strong surface winds had ushered in from the inland continental region more aerosols with small particle sizes, which were not optically sensitive. For all cases representing new particle formation and growth considered

in this study, a large contribution of organics to small particles was observed, which then decreased during the particle growth period, indicating that aerosol particles with small sizes generally contribute more organics to the total aerosol mass.

Under low AI conditions, LWP and DER significantly increase as LTS increased, but under high AI conditions, LWP and DER changed little. Differences in LWP and DER between low

and high LTS conditions were larger under low pollution conditions than under high pollution conditions. This suggests that the dependence of cloud properties is weakened due to the aerosol perturbation. The reduction in LWP and DER was greater in stable environments than in unstable environments, indicating that clouds in stable environments are more influenced by aerosol perturbations than those in more unstable regimes. DER significantly increased with

increasing LWP under low aerosol conditions, but increased slowly as LWP increased under high polluted conditions. Under high AI conditions, COD sharply increased with increasing LWP, but under low AI conditions, the increase was slower. It indicated that aerosols can influence on the pattern of between the cloud properties each other.

Analyses of the PDFs of COD, LWP, and DER for low and high aerosol loading conditions

in air mass cluster I and II suggests that  high aerosol loading can significantly make the shift in COD towards larger values, and in LWP and DER towards smaller values, and narrow the distribution of LWP and DER. The magnitude of FIE estimated under continental air-mass conditions ranged from 0.07±0.03 to 0.26±0.09 with a mean value of 0.16±0.03 and showed an increase trend as LWP increased. Magnitude of the FIE estimated for aerosols with a low mass

of organics were larger than those for aerosols with a high mass of organics. This suggests that clouds over regions dominated by aerosol particles containing mostly inorganics are more



susceptible to aerosol perturbations, resulting in larger climate forcing, than clouds over regions dominated by aerosol particles containing mainly organics.

**Author contributions.** JL has performed the calculations and analyses of the research and wrote

the paper. ZL made the comments and participated in scientific discussion

**Competing interests.** The authors declare that they have no conflict of interest.

**Acknowledgements.** The ground-based measurements were obtained from the Atmospheric

Radiation Measurement (ARM) program sponsored by the U.S. Department of Energy (DOE) Office of Energy Research, Office of Health and Environmental Research, Environmental Sciences Division. The reanalysis data are obtained from the European Centre for Medium-Range Weather Forecasts (ECMWF) model runs for ARM provided by the ECMWF.

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





Table 1. Seasonal mean of aerosol properties during the campaign period.

|  | Spring | Summer | Autumn | Winter | Yearly |
|---|---|---|---|---|---|
| $\sigma_1$ (Mm$^{-1}$) | 14.2±14.1 | 33.7±28.0 | 14.4±13.6 | 12.8±11.7 | 18.1±19.3 |
| $\sigma_{10}$ (Mm$^{-1}$) | 31.2±25.3 | 45.0±32.9 | 26.5±20.4 | 26.3±23.6 | 31.7±26.7 |
| $\sigma_1 / \sigma_{10}$ | 0.455 | 0.749 | 0.543 | 0.487 | 0.568 |
| $N_a$ (m$^{-3}$) | 2868±2367 | 2498±1536 | 2280±1854 | 2611±2108 | 2559±2014 |
| SSA | 0.95±0.04 | 0.96±0.03 | 0.95±0.04 | 0.94±0.04 | 0.95±0.04 |
| AOD$_{440}$ | 0.11±0.08 | 0.19±0.14 | 0.11±0.11 | 0.08±0.05 | 0.13±0.1 |
| AE | 1.27±0.40 | 1.65±0.31 | 1.51±0.36 | 1.35±0.45 | 1.44±0.40 |

$\sigma_1$: scattering coefficient, fine-mode particles; $\sigma_{10}$: scattering coefficient, total; $N_a$: aerosol number concentration; SSA: single scattering albedo; AOD$_{440}$: aerosol optical depth at 440 nm; AE: Angstrom exponent



**Table 2.** Mean and standard deviation of aerosol and cloud properties for each cluster of air mass.

| Air Mass | $\sigma_1$ (Mm$^{-1}$) | $\sigma_{10}$ (Mm$^{-1}$) | $\sigma_1/\sigma_{10}$ | $N_a$ (m$^{-3}$) | SSA | AOD$_{440}$ | AE | COD | LWP (g m$^{-2}$) | DER (μm) |
|---|---|---|---|---|---|---|---|---|---|---|
| I | 19.8±21.4 | 30.6±26.8 | 0.65 | 2969±2183 | 0.94±0.04 | 0.13±0.12 | 1.6±0.4 | 25.7±14.5 | 127±99 | 7.9±4.8 |
| II | 14.5±14.6 | 30.8±25.6 | 0.47 | 1788±1322 | 0.96±0.04 | 0.11±0.08 | 1.3±0.5 | 33.0±18.3 | 243±197 | 10.9±6.6 |
| III | 16.4±15.6 | 34.5±27.2 | 0.48 | 1937±1558 | 0.96±0.03 | 0.12±0.06 | 1.3±0.4 | 26.4±16.5 | 162±121 | 9.8±4.9 |

$\sigma_1$: scattering coefficient, fine-mode particles; $\sigma_{10}$: scattering coefficient, total; $N_a$: aerosol number concentration; SSA: single scattering albedo;

AOD$_{440}$: aerosol optical depth at 440 nm; AE: Angstrom exponent;

COD: cloud optical depth; LWP: liquid water path; DER: cloud droplet effective radius





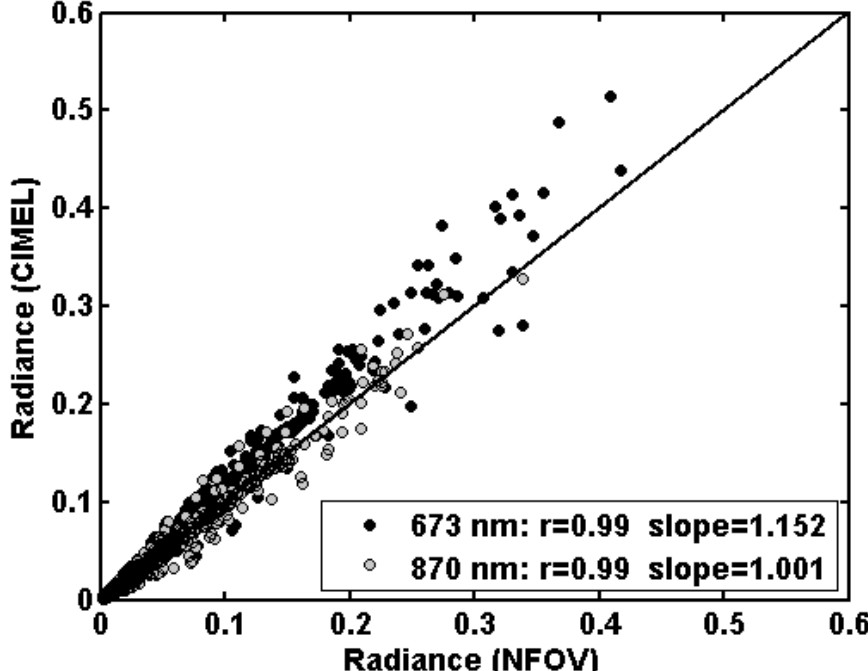

**Figure 1.** CIMEL sunphotometer-measured radiance as a function of narrow-field-of-view (NFOV) radiometer-measured radiance at 673 nm (black dots) and 870 nm (gray dots). The diagonal line represents the 1:1 line. Units are W sr$^{-1}$ m$^{-2}$.




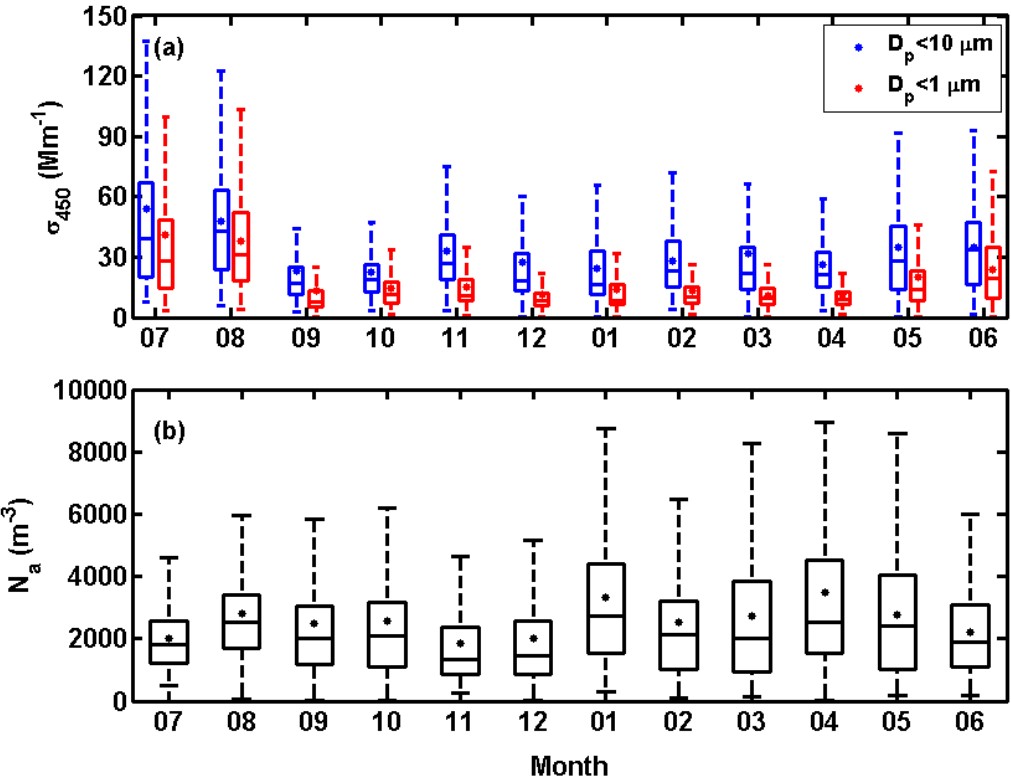

**Figure 2.** Monthly variations in (a) aerosol scattering coefficient at 450 nm ($\sigma_{450}$) for total (in blue) and fine mode (in red) aerosol particles and (b) aerosol particle number concentration ($N_a$). Box and whisker plots include median values (horizontal lines inside boxes), 25th and 75th percentiles (ends of boxes), 5th and 95th percentiles (ends of whiskers), and mean values (black dots). Months from left to right start at July 2012 and end at June 2013.





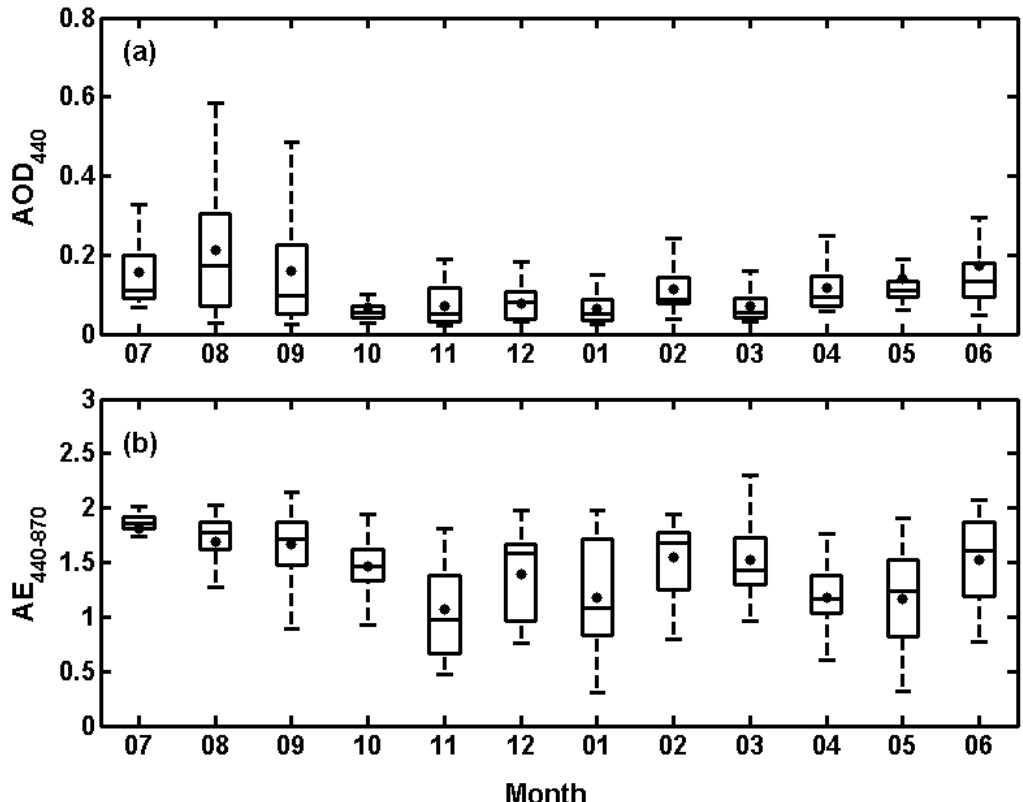

**Figure 3.** Monthly variations in columnar (a) aerosol optical depth at 440 nm ($AOD_{440}$) and (b) Angstrom exponent (AE). Box and whisker plots include median values (horizontal lines inside boxes), 25th and 75th percentiles (ends of boxes), 5th and 95th percentiles (ends of whiskers), and mean values (black dots). Months from left to right start at July 2012 and end at June 2013.




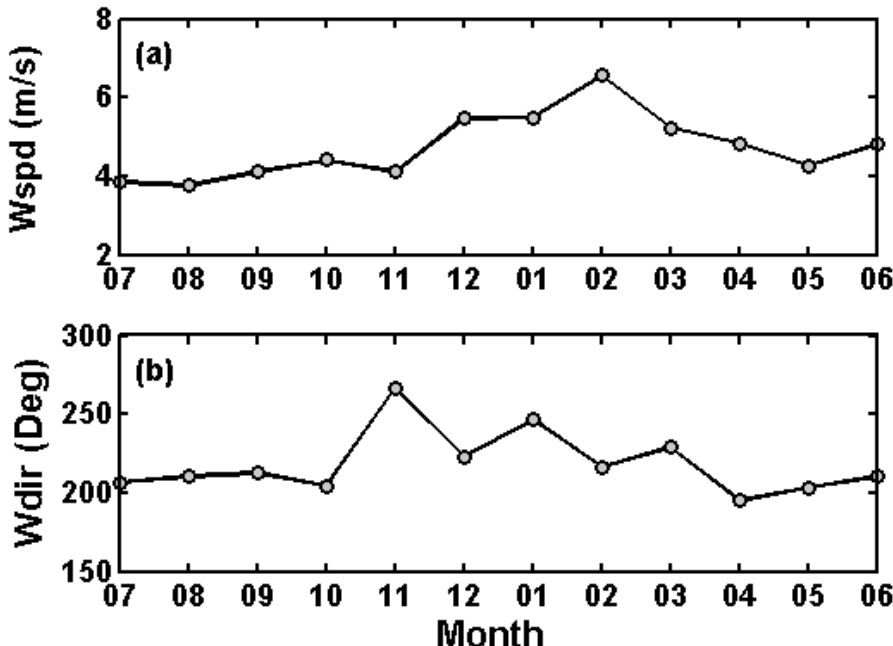

**Figure 4.** Monthly mean (a) wind speed (Wspd) and (b) wind direction (Wdir) during the campaign period. Months from left to right start at July 2012 and end at June 2013.





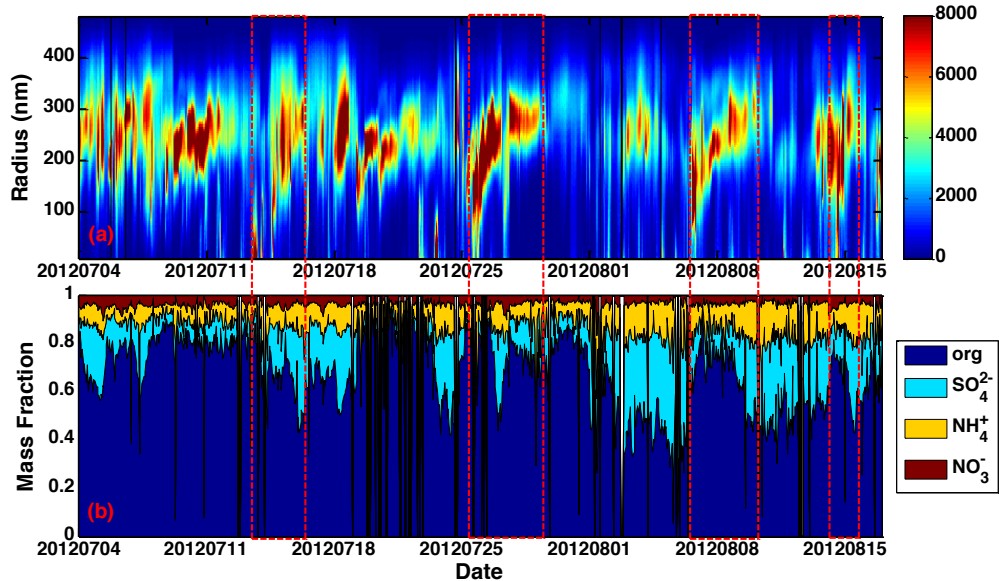

**Figure 5.** Time series of (a) particle size distribution and (b) mass fraction of organics (org, dark blue), sulfate ($SO_4^{2-}$, aqua), ammonium ($NH_4^+$, yellow), and nitrate ($NO_3^-$, red) in aerosols sampled during July and August of 2012. Dashed red rectangles outline periods of new particle formation and growth.




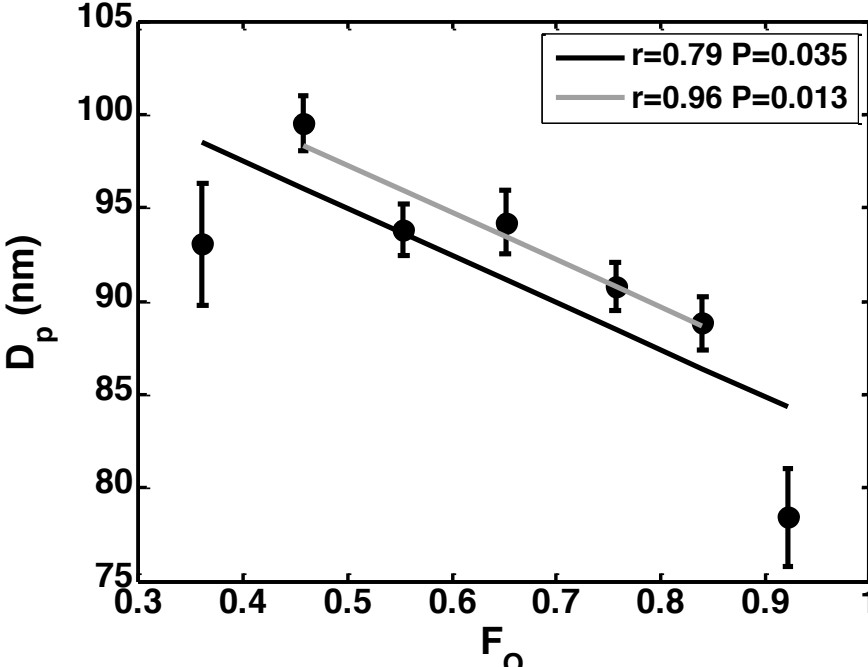

**Figure 6.** Mean aerosol particle radius ($D_p$) as a function of organic mass fraction ($F_O$). The black line is the linear regression line for all FO bins. The gray line is the linear regression line for $F_O$ bins ranging from 0.4 to 0.9, which have the most samples. Data are from July and August of 2012.




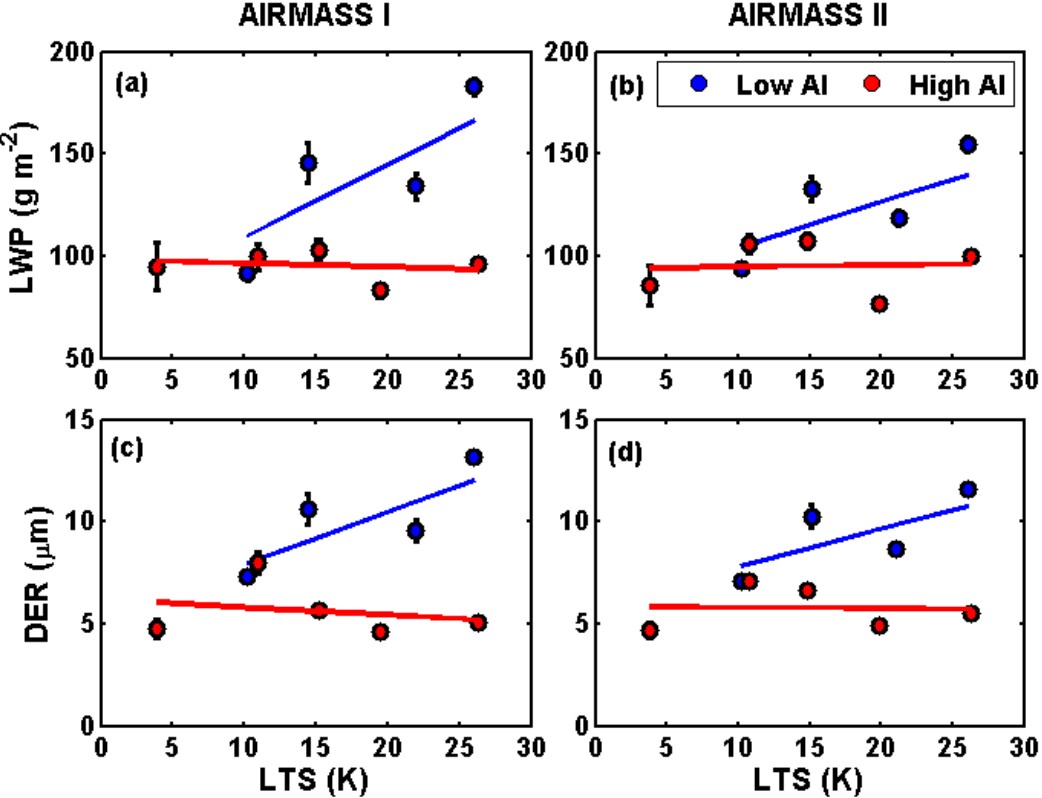

**Figure 7.** Liquid water path (LWP) and cloud droplet effective radius (DER) as functions of lower tropospheric stability (LTS) at low (in blue) and high (in red) aerosol index (AI) levels for cluster I air-mass (a, c) and cluster II air-mass (b, d) conditions, respectively. Low and high AI are defined as the lowest and highest quarter of all AI samples, respectively.



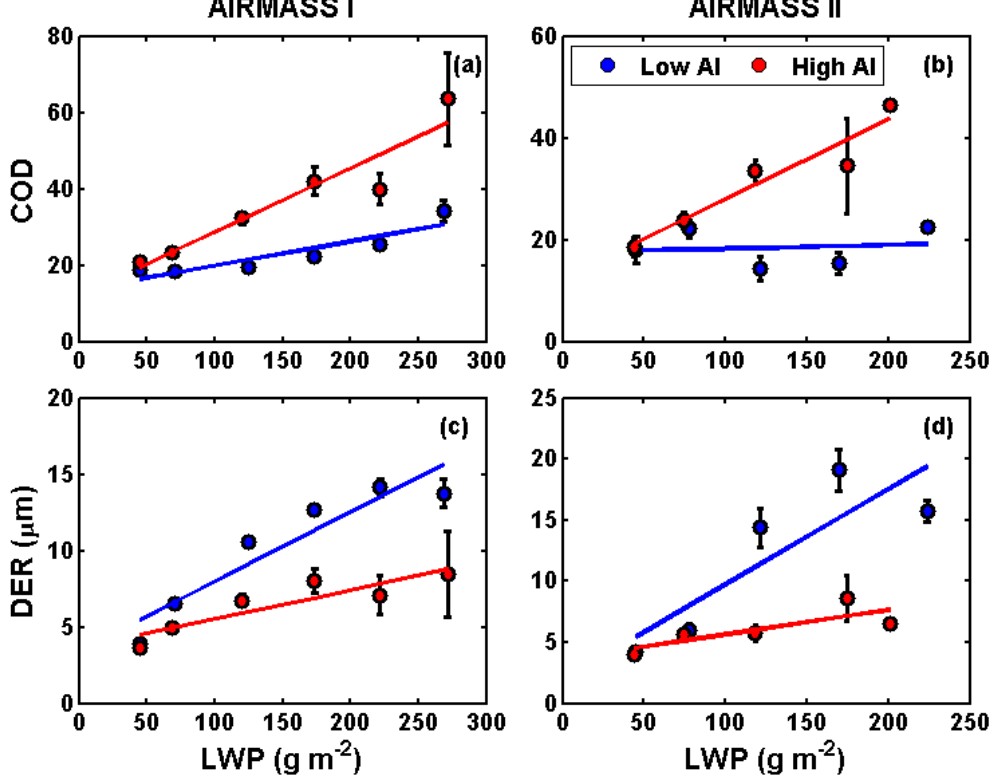

**Figure 8.** Cloud optical depth (COD) and DER as a function of liquid water path (LWP) at low (in blue) and high (in red) aerosol index (AI) levels for cluster I air-mass (a, c) and cluster II air-mass (b, d) conditions, respectively. Low and high AI are defined as the lowest and highest quarter of all AI samples, respectively.



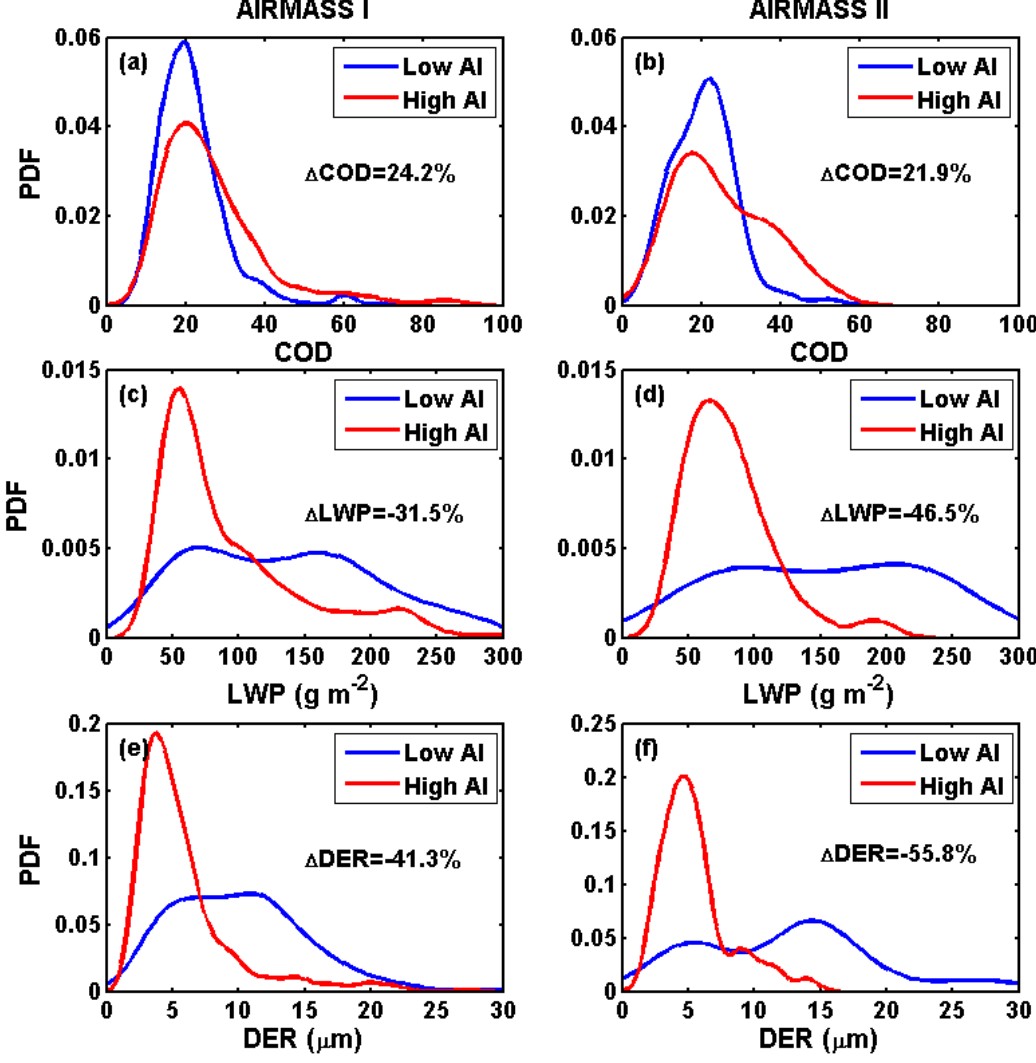

**Figure 9.** From top to bottom, probability distribution functions (PDF) of cloud optical depth (COD), liquid water path (LWP), and cloud droplet effective radius (DER) at low (in blue) and high (H, in red) aerosol index (AI) levels for cluster I air-mass (a, c, e) and cluster II air-mass (b, d, f).





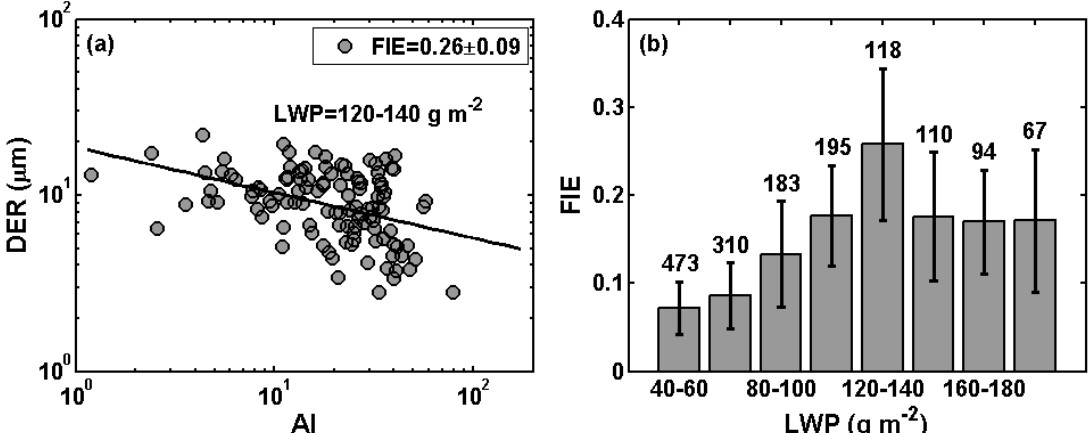

**Figure 10.** (a) Cloud droplet effective radius as a function of aerosol index (AI) for a sample bin with a constant liquid water path (LWP) range equal to 120-140 g m$^{-2}$, (b) the quantified aerosol first indirect effect (FIE) for each LWP bin. Numbers above each bar in (b) are the number of samples that went into the calculation of the FIE.





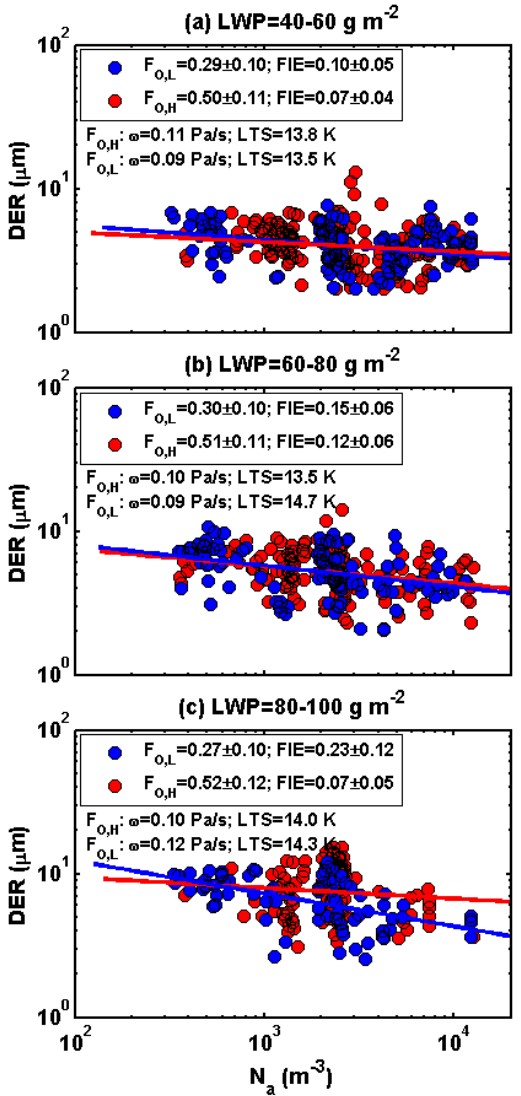

**Figure 11.** Cloud droplet effective radius (DER) as function of aerosol number concentration ($N_a$) at low (in blue) and high (in red) levels of mass fraction of organics in three liquid water path (LWP) bins: (a) 40-60 g m$^{-2}$, (b) 60-80 g m$^{-2}$, and (c) 80-100 g m$^{-2}$. Linear regression lines through each set of data are drawn. $F_{o,l}$ and $F_{o,h}$ are defined as the means of values less than and greater than, respectively, the mean value of the mass fraction of organics from all samples in each LWP bin. Mean value of $F_{o,l}$ and $F_{o,h}$ with their standard deviations and magnitude of FIE with their uncertainties are given in the legends. Mean value of vertical velocity (ω), and lower tropospheric stability (LTS) corresponding to $F_{o,l}$ and $F_{o,h}$ levels at each LWP bin are also shown.