# Peer review of "Aerosol Properties and Their Influences on Low Warm Clouds during the Two-Column Aerosol Project"

_Atmospheric Chemistry and Physics, 2019_

## Referee Comment (RC1) · Anonymous Referee #1 · 25 Mar 2019

Comments to the Authors Estimating the influence of aerosol on cloud properties is an important issue, which makes the impact of aerosols on the prediction of current and future behavior of earth's climate system highly uncertain. The manuscript entitled "Aerosol Properties and Their Influences on Low Warm Clouds during the Two-Column Aerosol Project" illustrated the analysis on the aerosol physical, optical, and chemical properties, and their influence on the dependence of cloud development on thermo-dynamic based on Twelve months of measurements collected during the Two-Column Aerosol Project field campaign over Cape Cod, Massachusetts. The manuscript has been well organized and properly addressed, as well as interesting results are found. Therefore, the paper could be considered to be published after minor Revision. Several

concerns of the manuscript are: 1. Page 6, Line 23, "700 hPa and LTS is......" should be "700 hPa and LTS are......". 2. Page 7, Line 13, "Figure 2 shows......" should be "Figure 2 shows......". 3. What's the meaning of "P" in Figure 6? 4. Page 8, Line 4-5, "This indicates that strong surface wind speeds transported smaller aerosol particles with no optical sensitivity from the continental interior to over the site", how to get the conclusion that the smaller aerosols are transported from the continental interior? 5. Why do the periods shown in red box in Figure 5 represent the periods of new particle formation and growth? 6. Page 9, Line 15. Actually, as shown in Nakajima et al (2001) and Liu and Li (2014), aerosol index is defined as the product of AOD and AE. However, in this study, aerosol index is defined as the product of the surface-measured aerosol scattering coefficients and AE in this study? 7. The linear regression slopes need be given in Figure 7. 8. Page 10, Line 30, "Figure 8c and 8d suggests ..." should be "Figure 8c and 8d suggest ..." 9. Page 11, Line 6, "Figures 8 ..." should be "Figure 8 ..." 10. Page 11, Line 10, "... suggests that ..." should be "... suggest that ..." 11. What is the lack of samples for cluster II air-mass condition? Why cannot use the cluster II to calculate FIE? 12. Page 13, Line 6, "... can is possibly because ..." should be "... can be possibly because ..." 13. Page 14, Line 12-15, "... more inorganic compounds that when ..." should be "... more inorganic compounds than that when ..."

---

## Referee Comment (RC2) · Anonymous Referee #2 · 25 Mar 2019

Liu and Li "Aerosol Properties and Their Influences on Low Warm Clouds during the Two-Column Aerosol Project"

Based on field experiement observations from Two-Column Aerosol Project field campaign over Cape Cod, Massachusetts, this study investigates the aerosol properties and their impacts on the cloud development. Aerosol first indirect effect (FIE) is quantified and its sensitivities to aerosol compositions are examined. While there are a lot of FIE quantification studies, the uncertainties related to aerosol first indirect effect are still large and theoretical understandings are still needed. The findings of this study could help advance our knowledge in the field of aerosol-cloud interaction. Thus, I

would recommend its acceptance for publication after a minor revision.

Page 1. Line 26-29, please modify the sentence to correct the grammar. Page 1, Line 30, either "a stable condition" or "stable conditions" Page 2. line 2, you may remove "with increasing LWP" Page 2, Line 3-5, grammar error for "can significantly made", also "narrowed" Page 2, Line 15-28, many references should be cited here. For example, the aerosol direct effect (Yang et al. 2016, 2018; doi:10.1002/2016JD024938, doi: 10.1016/j.atmosres.2018.04.029), the aerosol indirect effect (Feingold et al., 2003; Garrett et al., 2004; Zhao et al., 2018, 2019), lifetime effect (Albrecht 1989), thermal emissivity effect (Zhao and Garrett, 2006; Garrett and Zhao, 2015); semidirect effect (Koren et al.). Feingold, et al., First measurements of the Twomey indirect effect using ground-based remote sensors, Geophys. Res. Lett., 30(6), 1287, doi:10.1029/2002GL016633, 2003. Garrett, et al., 2004: Effects of varying aerosol regimes on low-level Arctic stratus. Geophys. Res. Lett., 31, L17105. Zhao, et al. (2018). Negative Aerosol-Cloud re Relationship from Aircraft Observations over Hebei, China. Earth and Space Science, 5, 19-29. Zhao, et al. (2019), A case study of stratus cloud properties using in situ aircraft observations over Huanghua, China, Atmosphere, 10, 19. Albrecht, B.A., 1989: Aerosols, cloud microphysics, and fractional cloudiness, Science, 245(4923), 1227-1230. Garrett, T. J. and C. Zhao, 2006: Increased Arctic cloud longwave emissivity associated with pollution from mid-latitudes. Nature, 440, nature04636, 787-789. Zhao, C., and T. Garrett, 2015:ÂăEffects of Arctic haze on surface cloud radiative forcing,ÂăGeophys. Res. Lett.,Âă42, 557-564, doi:10.1002/2014GL062015. Page 2, Line 32, for surface remote sensing, Garrett et al. (2004) and Qiu et al. (2017, 8-Year ground-based observational analysis about the seasonal variation of the aerosol-cloud droplet effective radius relationship at SGP site) should be cited. Page 3, Line 2, for aircraft measurement-based studies, Yang et al. (2019, Toward understanding the process-level impacts of aerosols on microphysical properties of shallow cumulus cloud using aircraft observations) and Zhao et al. (2018, 2019) should be cited, which are for North China region. Page 3. Line 17-20, The effect is also dependent on the availability of water vapor, or the amount of water vapor,

and meteorology (such as vertical velocity), as indicated by Qiu et al. (2017) and Yang et al. (2019). Page 3, Line 22-23, Garrett et al. (2004) also examined the sensitivity of FIE to aerosol size and number, which shows weak sensitivity of FIE to aerosol number concentration for those small sizes, but good sensitivity for aerosols with relatively large size (such as CCN or accumulation mode aerosol). Page 4. Line 13-14, What is the maximum size for Na? You might also give this information. Page 5 Line 10-21, The uncertainty information for cloud boundaries should be provided. As indicated by Zhao et al. (2012, Toward Understanding of Differences in Current Cloud Retrievals of ARM Ground-based Measurements) and Zhao et al. (2013, Ground-based remote sensing of thin clouds in the Arctic), the uncertainties in cloud bases and tops measured by ARM are generally 7.5 m and 45 m, respectively. Page 5. Line 28, "density of liquid water", and COD is cloud optical depth at visible wavelength. Page 8, Line 12-15, please check and correct the grammar here. Page 10, Line 1-5, Other studies as mentioned earlier have also indicated this likely evaporation and entrainment effect near cloud tops, which could be cited. Page 10, Line 7, "can possible" -> "can be possible" Page 10, Line 10-14, Yang et al. (2019), Zhao et al. (2018, 2019) have also made similar descriptions. Page 12, Line 20-21, Zhao et al. (2012) have indicated that using different aerosol variables to represent the aerosol loading amount, the quantified FIE values could vary, which is worthy to be mentioned here. Page 13, Line 11, also Zhao et al. (2012); Lin 12-13, Yang et al. (2019) too. Page 13, Line 24, I would suggest "the question how sensitive the cloud properties are sensitive to ..." Page 13, Line 26, what is the size range for the aerosol concentration? Page 14, Line 13, "larger"?

---

## Short Comment (SC1) · 20 Apr 2019

This study investigates the aerosol properties and sensitivity of clouds to aerosol perturbation at Cape Cod, Massachusetts. During a 12-month period, the physical, chemical and optical properties of aerosol are investigated under both continental and marine air masses. The aerosol effects on cloud properties are also examined, which demonstrates different responses of cloud properties to meteorological factors under low and high amount of aerosol penetration. The aerosol first indirect effect is quantified under the influence of continental air masses, with the magnitude generally consistent with previous studies. An interesting result influencing aerosol composition on cloud micro-

physical responses is also shown in this study, which adds on the information of FIE assessments and broaden the understanding of aerosol cloud interaction. Therefore, I would suggest that this paper can be accepted by ACP after minor revision.

Questions and concerns regarding this study are listed as follow:

1. Page 4, Line 23. What is the temporal resolution of Na and aerosol optical properties used in this study?

2. Page 6, Line 25. LTS is calculated at 1-min resolution?

3. Page 6, Line 29. Please specify how to collocate the datasets of different time resolutions (e.g. ACSM, AOS, LTS, large-scale vertical velocity and cloud properties) for the comparisons later on in the manuscript, particularly shown in Figure 10 & 11. And the final temporal resolution for collocated data.

4. Page 7, Line 23-25. For Spring season, AE values generally lower than other season, especially for April and May as shown in Figure 3 and Table 1. Also, $\sigma 1/\sigma 10$ value is lowest in Spring. Which indicates aerosol plumes more enriched by larger particles, relatively. Please provide more evidences or paper citations to support the statement "due to the presence of a great number of smaller particles...".

5. Page 8, Line 2-6. Please specify the exact months in this argument, and how to conclude that "This indicates that strong surface..."

6. Page 8, Line 25. Why only data of July and August 2012 are shown? How about particle size distribution in Spring and Autumn, since they are argued in section 3.1.1 as having discrepancies between Na and $\sigma$ due to particle size distributions.

7. Page 9, Line 1. Please specify the bin sizes used for low and high AI condition. Is there any reason for the mismatched bins between those two conditions, as shown in Figure 7?

8. Page 10, Line 8. "positive correlation each other" should be "positive correlation

between each other".

9. Page 11, Line 30. "a narrower PDF a distinct peak" should be "a narrower PDF with distinct peak".

10. Page 12, Line 23-24. How about FIE under cluster III which has occurrence of 21.6%, and how to determine samples are not enough under cluster II.

11. Page 13, Line 5-9. For ground-based assessments of FIE, Kim et al. (2008) and McComiskey et al. (2009) found decrease of FIE with LWP due to enhanced collision coalescence, please provide the information of cloud droplet number concentration to support the statement "more droplets can get activated".

---

## Author Comment (AC1) · 18 Jun 2019

**Response to Reviewer #1:**

The authors of the present manuscript acknowledge the reviewer for carefully reading and providing constructive comments that have led to an improved paper. Responses are written in blue text.

1. Page 6, Line 23, "700 hPa and LTS is. . .. . ." should be "700 hPa and LTS are......".

**Response:** Done.

2. Page 7, Line 13, "Figure 2 shows......" should be "Figure 2 shows. . .. . .".

**Response:** Done.

3. What's the meaning of "P" in Figure 6?

**Response:** P is the statistical probability. This information has been added to Figure 6's caption.

4. Page 8, Line 4-5, "This indicates that strong surface wind speeds transported smaller aerosol particles with no optical sensitivity from the continental interior to over the site", how to get the conclusion that the smaller aerosols are transported from the continental interior?

**Response:** We found that this statement is not fully supported by the current dataset and analysis. We have thus removed the sentences "However, relatively large aerosol number concentrations were measured. This indicates that strong surface wind speeds transported smaller aerosol particles with no optical sensitivity from the continental interior to over the site." from the revised manuscript.

5. Why do the periods shown in red box in Figure 5 represent the periods of new particle formation and growth?

**Response:** From an observational point of view, atmospheric new particle formation and subsequent particle growth are seen as the emergence of new aerosol particles into the lower end of the measured particle size spectrum (e.g., particle sizes below 50 nm), followed by the growth of these particles into larger sizes (Kulmala et al., 2012). The periods outlined in red show that aerosol particles start off small then grow larger.

[ Kulmala, M., et al., 2012. Measurement of the nucleation of atmospheric aerosol particles. Nat. Protoc. 7, 1651–1667.]

6. Page 9, Line 15. Actually, as shown in Nakajima et al (2001) and Liu and Li (2014), aerosol index is defined as the product of AOD and AE. However, in this study, aerosol index is defined as the product of the surface-measured aerosol scattering coefficients and AE in this study?

**Response:** We have changed the term "aerosol index" to "scattering aerosol index", which has been used in related studies (e.g., Liu and Li, 2014; Sena et al., 2016). We have also deleted the reference to Nakajima et al. (2001) because they define the scattering aerosol index differently.

[Liu, J., and Li, Z.: Estimation of cloud condensation nuclei concentration from aerosol optical quantities: influential factors and uncertainties, Atmos. Chem. Phys., 14(1), 471–483, https://doi.org/10.5194/acp-14-471-2014, 2014.

Sena, E. T., McComiskey, A., and Feingold, G.: A long-term study of aerosol-cloud interactions and their radiative effect at the Southern Great Plains using ground-based measurements, Atmos. Chem. Phys., 16, 11,301–11,318, doi:10.5194/acp-16- 11301-2016, 2016.]

7. The linear regression slopes need be given in Figure 7.

**Response**: The slopes have now been given in Figures 7 and 8.

8. Page 10, Line 30, "Figure 8c and 8d suggests ..." should be "Figure 8c and 8d suggest ..."

**Response**: Done.

9. Page 11, Line 6, "Figures 8 ..." should be "Figure 8 ..."

**Response**: Done.

10. Page 11, Line 10, "... suggests that ..." should be "... suggest that ..."

**Response**: Done.

11. What is the lack of samples for cluster II air-mass condition? Why cannot use the cluster II to calculate FIE?

**Response**: As shown in Figure 10, combined cloud and aerosol data need to be separated into narrow LWP bins to calculate the FIE. There were not enough cloud and aerosol samples in each narrow LWP bin for the cluster II air mass to avoid large uncertainties in the FIE estimates. Furthermore, only cases with sample numbers greater than 50 and with calculated values of FIE that are statistically significant at the 95% confidence level (P = 0.05) are discussed in the study. To make things clearer, we have changed the sentence "Due to the lack of samples …" to "Since there were not enough samples under cluster II air-mass condition …". We also added the sentence "Only those cases with sample numbers greater than 50 per bin and where the calculated values of FIE are statistically significant at the 95% confidence level (P = 0.05) are analyzed here." to the revised manuscript.

12. Page 13, Line 6, "... can is possibly because ..." should be "... can be possibly because ..."

**Response**: Done.

13. Page 14, Line 12-15, "... more inorganic compounds that when ..." should be "... more inorganic compounds than that when ..."

**Response**: Sentence corrected.

---

## Author Comment (AC2) · 18 Jun 2019

**Response to Reviewer #3**

The authors of the present manuscript acknowledge the reviewers for carefully reading and providing constructive comments that have led to an improved paper. Responses are written in blue text.

1. Page 4, Line 23. What is the temporal resolution of Na and aerosol optical properties used in this study?

**Response:** The temporal resolution is one minute. This information has been added to section 2.1.1: "The time resolution of the $N_a$, $\sigma_s$, and $\sigma_a$ measurements is one minute."

2. Page 6, Line 25. LTS is calculated at 1-min resolution?

**Response**: The temporal resolution is one hour. This information has been added to section 2.3: "The European Centre for Medium-Range Weather Forecasts model runs for ARM analysis with a one-hour resolution for a 0.56º x 0.56º box centered on the site provided values of ω and potential temperature."

3. Page 6, Line 29. Please specify how to collocate the datasets of different time resolutions (e.g. ACSM, AOS, LTS, large-scale vertical velocity and cloud properties) for the comparisons later on in the manuscript, particularly shown in Figure 10 & 11. And the final temporal resolution for collocated data.

**Response:** Most of the datasets, i.e., aerosol properties (scattering coefficients and number concentrations) from the AOS, cloud properties (COD, LWP, DER), and surface meteorology, have a 1-min temporal resolution. These data were first matched according to the observation time, and then matched with aerosol composition measurements and ECMWF simulations (LTS and large-scale vertical velocity) and integrated over 1-min time intervals. This means that the aerosol composition and LTS (vertical velocity) in the 1-min resolution datasets are assumed to remain constant within 30-min and 1-hour time periods, respectively.

We have added the sentence "To investigate the influence of aerosols on cloud properties, aerosol properties ($N_a$, $\sigma_s$, composition), cloud properties (COD, LWP, DER, boundary-layer height), surface meteorological parameters, and ECMWF simulations (LTS, large-scale vertical velocity) were matched according to observation time and averaged and interpolated over 1-min time intervals. " to the revised manuscript.

4. Page 7, Line 23-25. For Spring season, AE values generally lower than other season, especially for April and May as shown in Figure 3 and Table 1. Also, σ1/σ10 value is lowest in Spring. Which indicates aerosol plumes more enriched by larger particles, relatively. Please provide more evidences or paper citations to support the statement "due to the presence of a great number of smaller particles. . .".

**Response:** We found that this statement is not fully supported by the current dataset and analysis. We have thus removed the sentence "The largest values of $N_a$ corresponding to moderate values of $\sigma_s$ are found in spring and are likely due to the presence of a greater number of smaller particles with less optical sensitivity."

5. Page 8, Line 2-6. Please specify the exact months in this argument, and how to conclude that "This indicates that strong surface. . ."

**Response:** We have added the specific months to the sentence: "Months in summer and winter with the strongest mean surface wind speeds (e.g., June and January/February, respectively) …". The next sentence is not fully supported by the current dataset and analysis, so we have removed the "This indicates that strong surface …" sentence.

6. Page 8, Line 25. Why only data of July and August 2012 are shown? How about particle size distribution in Spring and Autumn, since they are argued in section 3.1.1 as having discrepancies between Na and σ due to particle size distributions.

**Response:** Here, we intended to examine the relationship between particle size and particle chemical composition. However, data were not available in spring and autumn.

7. Page 9, Line 1. Please specify the bin sizes used for low and high AI condition. Is there any reason for the mismatched bins between those two conditions, as shown in Figure 7?

**Response:** We have added the sentence "The cloud properties were averaged over each 6-K LTS bin from 0 K to 30 K under low and high scattering AI conditions."
The x-axis represents the mean values of LTS in each LTS bin. This is the reason for the mismatched bins between the two conditions.

8. Page 10, Line 8. "positive correlation each other" should be "positive correlation between each other".

**Response:** The sentence has been changed to "The changes in DER with LTS possibly reflect the changes in LWP with LTS due to the high positive correlation between LWP and DER (Zhang et al., 2011; Sporre et al., 2014)."

9. Page 11, Line 30. "a narrower PDF a distinct peak" should be "a narrower PDF with distinct peak".

**Response:** The sentence has been changed to "The high aerosol loading cases, conversely, have narrower PDFs with distinct peaks between 60 and 70 g m$^{-2}$."

10. Page 12, Line 23-24. How about FIE under cluster III which has occurrence of 21.6%, and how to determine samples are not enough under cluster II.

**Response:** The air-mass clusters were determined for each day during the observation period. Although 15.9% and 21.6% of the daily trajectories belong to the clusters II and III, the number of combined cloud and aerosol samples passing the screening criteria (as described in section 2.2.2) are insufficient to do such an analysis. Furthermore, only cases with sample numbers greater than 50 and with calculated values of FIE that are statistically significant at the 95% confidence level (P = 0.05) are discussed in the study.
We have added the sentence "Only those cases with sample numbers greater than 50 per bin and where the calculated values of FIE are statistically significant at the 95% confidence level (P = 0.05) are analyzed here." to the revised manuscript.

11. Page 13, Line 5-9. For ground-based assessments of FIE, Kim et al. (2008) and McComiskey et al. (2009) found decrease of FIE with LWP due to enhanced collision coalescence, please provide

the information of cloud droplet number concentration to support the statement "more droplets can get activated".

**Response**: Conflicting findings regarding the dependence of the FIE on the LWP have been reported, i.e., a positive correlation in some studies (Pandithurai et al., 2009; Harikishan et al., 2016), a negative correlation in others (Kim et al. 2008; McComiskey et al., 2009; Liu et al., 2016), and an independence of the FIE on the LWP (Lihavainen et al., 2010; Zhao et al., 2012). Different mechanisms have been reported explaining the negative correlation (i.e., a decrease in cloud droplets due to enhanced collision-coalescence) and the positive correlation (i.e., an increase in cloud droplets due to enhanced aerosol activation). Thus, the dependence of the FIE on the LWP likely depends on which mechanism dominates during the study period in question.

Unfortunately, cloud droplet number concentrations were not available during the field campaign, but our results are consistent with some previous studies. The latter mechanism mentioned above possibly plays a dominant role.

---

## Author Comment (AC3) · 18 Jun 2019

**Response to Reviewer #2**

The authors of the present manuscript acknowledge the reviewer for carefully reading and providing constructive comments that have led to an improved paper. Responses are written in blue text.

Page 1. Line 26-29, please modify the sentence to correct the grammar.

**Response:** The sentence has been changed to "Under low aerosol loading conditions, the liquid water path (LWP) and droplet effective radius (DER) significantly increased with increasing LTS, but under high aerosol loading conditions, LWP and DER changed little, indicating that aerosols significantly weakened the dependence of cloud development on LTS."

Page 1, Line 30, either "a stable condition" or "stable conditions"

**Response**: Fixed.

Page 2. line 2, you may remove "with increasing LWP"

**Response**: We have removed "with increasing LWP".

Page 2, Line 3-5, grammar error for "can significantly made", also "narrowed"

**Response:** The sentence has been changed to "Under both continental and marine air-mass conditions, high aerosol loading can significantly shift COD towards larger values and LWP and DER towards smaller values, narrowing the distributions of LWP and DER."

Page 2, Line 15-28, many references should be cited here. For example, the aerosol direct effect (Yang et al. 2016, 2018; doi:10.1002/2016JD024938, doi: 10.1016/j.atmosres.2018.04.029), the aerosol indirect effect (Feingold et al., 2003; Garrett et al., 2004; Zhao et al., 2018, 2019), lifetime effect (Albrecht 1989), thermal emissivity effect (Zhao and Garrett, 2006; Garrett and Zhao, 2015); semi- direct effect (Koren et al.). Feingold, et al., First measurements of the Twomey indi- rect effect using ground-based remote sensors, Geophys. Res. Lett., 30(6), 1287, doi:10.1029/2002GL016633, 2003. Garrett, et al., 2004: Effects of varying aerosol regimes on low-level Arctic stratus. Geophys. Res. Lett., 31, L17105. Zhao, et al. (2018). Negative Aerosol-Cloud re Relationship from Aircraft Observations over Hebei, China. Earth and Space Science, 5, 19-29. Zhao, et al. (2019), A case study of stratus cloud properties using in situ aircraft observations over Huanghua, China, Atmosphere, 10, 19. Albrecht, B.A., 1989: Aerosols, cloud microphysics, and fractional cloudiness, Science, 245(4923), 1227-1230. Garrett, T. J. and C. Zhao, 2006: In- creased Arctic cloud longwave emissivity associated with pollution from mid-latitudes. Nature, 440, nature04636, 787-789. Zhao, C., and T. Garrett, 2015: Effects of Arctic haze on surface cloud radiative forcing, Geophys. Res. Lett., 42, 557-564, doi:10.1002/2014GL062015.

**Response**: The suggested references have been added to the revised manuscript.

Page 2, Line 32, for surface remote sensing, Garrett et al. (2004) and Qiu et al. (2017, 8-Year ground-based observational analysis about the seasonal variation of the aerosol-cloud droplet effective radius relationship at SGP site) should be cited.

**Response**: They have now been cited.

Page 3, Line 2, for aircraft measurement-based studies, Yang et al. (2019, Toward understanding the process-level impacts of aerosols on microphysical properties of shallow cumulus cloud using aircraft observations) and Zhao et al. (2018, 2019) should be cited, which are for North China region.

**Response**: They have now been cited.

Page 3. Line 17-20, The effect is also dependent on the availability of water vapor, or the amount of water vapor, and meteorology (such as vertical velocity), as indicated by Qiu et al. (2017) and Yang et al. (2019).

**Response**: Yes, we agree. The suggested references have now been cited.

Page 3, Line 22-23, Garrett et al. (2004) also examined the sensitivity of FIE to aerosol size and number, which shows weak sensitivity of FIE to aerosol number concentration for those small sizes, but good sensitivity for aerosols with relatively large size (such as CCN or accumulation mode aerosol).

**Response**: We added the sentence "Garrett et al. (2004) indicated a weak sensitivity of FIE to aerosols with small particle sizes but a stronger sensitivity to aerosols with relatively large sizes." to the revised manuscript.

Page 4. Line 13-14, What is the maximum size for Na? You might also give this information.

**Response:** We have added this information and changed the sentence to "with diameters larger than 10 nm and smaller than 3 μm".

Page 5 Line 10-21, The uncertainty information for cloud boundaries should be provided. As indicated by Zhao et al. (2012, Toward Understanding of Differences in Current Cloud Retrievals of ARM Ground-based Measurements) and Zhao et al. (2013, Ground-based remote sensing of thin clouds in the Arctic), the uncertainties in cloud bases and tops measured by ARM are generally 7.5 m and 45 m, respectively.

**Response:** We have added the sentence "The cloud-base and cloud-top height uncertainties are ~7.5 m and ~45 m, respectively (Zhao et al., 2012a; Garrett and Zhao, 2013)." to the revised manuscript.

Page 5. Line 28, "density of liquid water", and COD is cloud optical depth at visible wavelength.

**Response:** Done.

Page 8, Line 12-15, please check and correct the grammar here.

**Response:** The sentence has been changed to "When the continental air-mass influenced the site, fine particles dominated aerosol scattering and were responsible for ~65% of the total particle scattering, indicating that more anthropogenic aerosols with small particle sizes were transported to the site from continental regions to the west."

Page 10, Line 1-5, Other studies as mentioned earlier have also indicated this likely evaporation and entrainment effect near cloud tops, which could be cited.

**Response:** Done.

Page 10, Line 7, "can possible" -> "can be possible"

**Response:** The sentence has been changed to "The changes in DER with LTS possibly reflect the changes in LWP with LTS due to the high positive correlation between LWP and DER (Zhang et al., 2011; Sporre et al., 2014)."

Page 10, Line 10-14, Yang et al. (2019), Zhao et al. (2018, 2019) have also made similar descriptions.

**Response:** These references are now cited.

Page 12, Line 20-21, Zhao et al. (2012) have indicated that using different aerosol variables to represent the aerosol loading amount, the quantified FIE values could vary, which is worthy to be mentioned here.

**Response**: It has already been mentioned in section 3.3.2.

Page 13, Line 11, also Zhao et al. (2012); Lin 12-13, Yang et al. (2019) too.

**Response**: These references are now cited.

Page 13, Line 24, I would suggest "the question how sensitive the cloud properties are sensitive to …"

**Response**: The sentence has been changed to "Examined next is the sensitivity of cloud properties to aerosol chemical composition represented by the mass fraction of organics."

Page 13, Line 26, what is the size range for the aerosol concentration?

**Response**: The size range for the aerosol concentration is 10 nm to 3 μm in diameter. This is mentioned in section 2.1.1.

Page 14, Line 13, "larger"?

**Response:** Done.